# Prognostic Relevance of Multi-Antigenic Myeloma-Specific T-Cell Assay in Patients with Monoclonal Gammopathies

**DOI:** 10.3390/cancers15030972

**Published:** 2023-02-03

**Authors:** Ivana Lagreca, Vincenzo Nasillo, Patrizia Barozzi, Ilaria Castelli, Sabrina Basso, Sara Castellano, Ambra Paolini, Monica Maccaferri, Elisabetta Colaci, Daniela Vallerini, Patrizia Natali, Daria Debbia, Tommaso Pirotti, Anna Maria Ottomano, Rossana Maffei, Francesca Bettelli, Davide Giusti, Andrea Messerotti, Andrea Gilioli, Valeria Pioli, Giovanna Leonardi, Fabio Forghieri, Paola Bresciani, Angela Cuoghi, Monica Morselli, Rossella Manfredini, Giuseppe Longo, Anna Candoni, Roberto Marasca, Leonardo Potenza, Enrico Tagliafico, Tommaso Trenti, Patrizia Comoli, Mario Luppi, Giovanni Riva

**Affiliations:** 1Section of Hematology, Department of Medical and Surgical Sciences, University of Modena and Reggio Emilia, AOU Modena, 41124 Modena, Italy; 2Diagnostic Hematology and Clinical Genomics, Department of Laboratory Medicine and Pathology, AUSL/AOU Modena, 41124 Modena, Italy; 3Pediatric Hematology/Oncology Unit and Cell Factory, Istituto di Ricovero e Cura a Carattere Scientifico (IRCCS) Policlinico San Matteo, 27100 Pavia, Italy; 4Centre for Regenerative Medicine “S. Ferrari”, University of Modena and Reggio Emilia, 41125 Modena, Italy; 5Department of Oncology and Hematology, AOU Modena, 41124 Modena, Italy

**Keywords:** MGUS, multiple myeloma, smoldering myeloma, immunity, T cells, ELISpot

## Abstract

**Simple Summary:**

Multiple myeloma (MM) is a hematologic cancer originating from underlying precursor disorders, called Monoclonal Gammopathy of Undetermined Significance (MGUS) and Smoldering Multiple Myeloma (SMM). In order to predict the risk of progression from MGUS/SMM to MM, well-known prognostic factors are currently used in the clinical management, but the contribute of myeloma-specific T-cell immunity still remains to be addressed in this context. In this work, we shed light on the relevant dynamics of MM-specific T cells in a long-term cohort of MGUS/SMM patients, in particular describing a new prognostic tool, namely *Multi-antigenic Myeloma-specific (MaMs)* T-cell assay, which may be useful to assess the presence of a protective MM-specific immunity, possibly representing a novel immunologic marker of effective disease control. Our work can be considered a promising proof-of-concept for future larger studies, aiming to further explore the prognostic impact of MaMs-based tests in the management of patients with monoclonal gammopathies.

**Abstract:**

Multiple Myeloma (MM) typically originates from underlying precursor conditions, known as Monoclonal Gammopathy of Undetermined Significance (MGUS) and Smoldering Multiple Myeloma (SMM). Validated risk factors, related to the main features of the clonal plasma cells, are employed in the current prognostic models to assess long-term probabilities of progression to MM. In addition, new prognostic immunologic parameters, measuring protective MM-specific T-cell responses, could help to identify patients with shorter time-to-progression. In this report, we described a novel Multi-antigenic Myeloma-specific (MaMs) T-cell assay, based on ELISpot technology, providing simultaneous evaluation of T-cell responses towards ten different MM-associated antigens. When performed during long-term follow-up (mean 28 months) of 33 patients with either MGUS or SMM, such deca-antigenic myeloma-specific immunoassay allowed to significantly distinguish between stable vs. progressive disease (*p* < 0.001), independently from the Mayo Clinic risk category. Here, we report the first clinical experience showing that a wide (multi-antigen), standardized (irrespective to patients’ HLA), MM-specific T-cell assay may routinely be applied, as a promising prognostic tool, during the follow-up of MGUS/SMM patients. Larger studies are needed to improve the antigenic panel and further explore the prognostic value of MaMs test in the risk assessment of patients with monoclonal gammopathies.

## 1. Introduction

Multiple myeloma (MM) is a hematologic malignancy characterized by the outgrowth of clonal plasma cells (PCs) in the bone marrow (BM), with distinctive secretion of a monoclonal immunoglobulin (M-protein), progressively leading to organ damage, typically manifesting as anemia, renal impairment, and bone fractures [1]. Virtually in all cases, MM originates from an underlying precursor disorder, known as Monoclonal Gammopathy of Undetermined Significance (MGUS), which may also evolve through an asymptomatic disease stage, defined as Smoldering Multiple Myeloma (SMM) [1,2,3,4]. The risk of progression from MGUS/SMM to active MM widely varies, depending on individual disease features, generally ranging from about 5% (MGUS) to 50% (SMM) within 5 years from the diagnosis [1]. Several risk factors, mainly related to the clonal PC disease burden, have been identified to distinguish patients with different long-term risk of MGUS/SMM-to-MM progression. The consolidated prognostic tools currently in use (namely, Mayo Clinic and PETHEMA risk assessment scores) are classically based on clone-associated main parameters, i.e., (i) the type of M-protein (typically, IgG or IgA), (ii) the amount of M-protein, (iii) the ratio between involved/uninvolved free light chains (FLCs), and (iv) the percentage of BM infiltration by clonal PCs. Interestingly, (v) the reduction of non-involved Ig levels (immunoparesis) is the sole prognostic factor regarding an immunological parameter, rather than an intrinsic clone feature. Of note, the current prognostic stratifications are validated to predict a long-term risk (5–20 years) of MGUS/SMM-to-MM evolution, but are less useful to predict a shorter time-to-progression (<5 years) [1,2,3,4,5]. Latest genomic evidence well highlighted that myeloma is not a single disease entity but rather a spectrum of clonally evolving PC proliferations, genetically heterogenous since precursors stages, eventually manifesting with similar clinical features [6]. Going beyond the validated risk stratification models, some important prognostic factors—able to early identify high risk patients at each disease stage—still remain elusive [7,8].

In healthy individuals with MGUS, as well as in asymptomatic patients with SMM, the ultimate goal for optimal clinical management should be the early identification of patients with rapidly progressive disease (i.e., showing a higher risk to develop symptomatic MM in few years) and offer them an intensified follow-up program, aimed to timely prevent the occurrence of end-organ damages, as well as, possibly, to guide “pre-emptive” therapeutic approaches in selected high-risk patients. On the other hand, such a prognostic improvement should also allow to plan more appropriate (less strict) follow-up schedules for subjects with stable gammopathy.

In the MGUS/SMM-to-MM progression, full-blown neoplastic proliferation of aberrant PCs requires the combination of interdependent processes of genomic instability, clonal evolution, and complex deregulations of BM microenvironment, in particular with pivotal impairment of antineoplastic T-cell surveillance, ultimately leading to a tumor-supporting inflammatory milieu and to the immune escape of clonal PCs [2,9,10,11,12]. Interestingly, the notion that the immune system can recognize malignant plasma cells in MM patients and in subjects with MGUS was initially revealed by pioneering works more than 20 years ago and has become fully evident during the last decade. Indeed, since first reports about the existence of anti-idiotype T-cell responses in MM [13,14,15], a growing number of tumor-associated antigens (TAAs), such as WT1, SOX-2, RHAMM, PRAME, and several cancer–testis antigens (namely, MAGE-A1/A2/A3/A6, MAGE-C1/C2, NY-ESO-1, and Melan-A/MART-1), have been discovered to be also expressed by MM cells (thus, called MM-associated antigens, MMAAs) and to induce detectable T-cell responses, which often showed significant killing capacity against neoplastic PCs in vitro, as well as clinical associations with improved disease control in different patients’ series [16]. To date, several research groups have consistently described the presence of either circulating or marrow-resident MMAA-specific T cells in independent series of MM patients, as well as in subjects with MGUS. Originally, Dhodapkar et al. [17] demonstrated that both CD4+ and CD8+ BM T cells from MGUS patients can display vigorous cytotoxic responses against autologous PCs in vitro, but these activities were not evident when BM T cells from active MM patients were tested, thus supporting the notion that increasing tumor burden may correlate with cytotoxic T-cell exhaustion. Further studies showed that myeloma-specific T cells are able to recognize and eliminate both MGUS and MM clonal PCs, although such protective T-cell immunity is often lost in patients progressing to MM. In some cases, CD4+ and CD8+ T cells directed towards MAGE-A1 or SOX-2 were found at higher frequencies in MGUS patients, showing significant associations with reduced risk of progression to MM, improved disease control, and longer survival [18,19]. Moreover, WT1-specific T-cell responses have been reported in a series of relapsed MM patients undergoing allogeneic hematopoietic stem cell transplant (allo-HSCT), demonstrating a clear correlation between the emergence of such WT1-specific cytotoxic T lymphocytes (CTLs) and the graft-versus-myeloma effect, particularly in patients treated with donor lymphocyte infusions (DLIs) [20]. Similarly, MM patients receiving allo-HSCT showed antibody response to BCMA after DLI infusions, remarkably contributing to graft-versus-myeloma response [21]. Of note, the role of myeloma-specific memory T-cell expansions and the priming of naïve T cells by dendritic cells have also been recognized in MM patients undergoing autologous HSCT (auto-HSCT), possibly promoted by a lymphodepleted and inflammatory microenvironment [22]. Interestingly, T-cell dependent myeloma control in the setting of auto-HSCT (i.e., autologous graft-versus-tumor effect against neoplastic PCs) has formally been demonstrated in Vk*MYC myeloma-bearing mice, showing protective CD8+ T cells with a distinct T-cell receptor (TCR) repertoire and higher clonotype overlap, compared to controls [23,24].

By considering all these premises, the development of specific immunological tests, able to significantly measure the anti-myeloma immunity and, possibly, to complement current disease-related risk models, may deserve further investigations. Thus, moving from the promising works mentioned above, in 2018 we started a clinico-immunological study for the screening, characterization, and monitoring of several MM-specific T-cell responses in the peripheral blood (PB) of patients with either MGUS or SMM, referring to our hospital services. Such explorative study was aimed to find out an MMAA-specific immunoassay which could be easily standardizable among MGUS/SMM patients, i.e., being irrespective to patients’ HLAs, as well as not requiring the identification of MMAA expression profile for each case (it would imply invasive BM sampling and laborious PC isolation and immunophenotyping). Our investigations were particularly aimed at evaluating medium/long-term correlations between such measurable T-cell immunity (as assessed by antigen-specific ELISpot assay) and clinical outcomes, possibly identifying specific T-cell profiles and/or dynamics with prognostic relevance in this setting. In parallel, we also tested the feasibility of expanding ex vivo such MM-specific T cells, and we characterized these cells, both phenotypically and functionally, to reveal their therapeutic cytotoxic potentials.

## 2. Materials and Methods

### 2.1. Patients and Samples

We enrolled a cohort of 33 consecutive adult patients (median age 65, range 26–85), including 22 MGUS and 11 SMM. Written informed consent was obtained in accordance with the Declaration of Helsinki, and the study was approved by the University of Modena and Reggio Emilia ethical committee (protocol 229/12 CE AVEN). All patients were risk-stratified according to the Mayo Clinic prognostic models for MGUS and SMM patients [5]. Moreover, all patients were classified into two groups, according to their disease course, as observed during the entire period of follow-up: those with a stable “not-stepping” disease vs. those showing a raising, “stepping” disease. In particular, progressive disease was defined on the basis of a documented step to the subsequent Mayo Clinic risk category, or a proven progression from MGUS to SMM, or from MGUS/SMM to MM, in the timeframe of the study (from patient’s enrollment to the last follow-up evaluation).

PB samples were collected at the time of patient’s enrollment, then at each subsequent follow-up visit (typically every 3–6 months, according to clinical practice). BM samples were not included in this study. M-protein level and serum Free Light Chain (FLC) ratio were assessed at each follow-up visit, in order to correlate MMAA-specific T-cell responses with measurable disease activity markers. Median follow-up from enrollment was 28 months (range 6–44). Patients’ clinical features are summarized in Table 1.

### 2.2. Multi-Antigenic MM-Specific Antigen Stimulation 

To assure a broad stimulation with myeloma-associated antigens, we selected the 10 tumor proteins that, to date, have been demonstrated to elicit a myeloma-specific T-cell response. In details, such 10 MMAAs are: Melanoma Antigen Recognized by T-cells (Melan-A/MART-1) [25], New York esophageal squamous cell carcinoma 1 (NY-ESO-1) [26,27], Wilms’ Tumor 1 (WT1) [20], Melanoma-associated antigen 1 (MAGE-A1) [18], human telomerase reverse transcriptase (hTERT) [28], SRY-Box Transcription Factor 2 (SOX-2) [19,29], Dickkopf-1 (DKK1) [30], Junctional Adhesion Molecule 1 (JAM-1) [31], Signaling Lymphocyte Activation Molecule Family 7 (SLAMF7) [32], and B-cell maturation antigen (BCMA) [21,33]. For each MMAA, pools of overlapping peptides (15mers with 11 aa overlap) covering the whole sequence were purchased from JPT Peptide Technologies (Melan-A/MART-1, NY-ESO-1, WT1, MAGE-A1, hTERT, and SOX-2, all commercially available) or custom synthesized by Genscript (DKK1, JAM-1, SLAMF7, and BCMA). All 10 peptide pools were used as MM-specific stimulation for the immunological assays performed in this study, either individually or as a comprehensive mixture.

### 2.3. Enzyme-Linked Immunospot (ELISpot) Assay

Screening and quantitation of MM-specific Interferon-gamma (IFNγ)-secreting T cells were performed by applying ELISpot assay, according to manufacturer’s instructions and protocols we previously described [34,35], on total 152 freshly isolated peripheral blood mononuclear cell (PBMC) samples, serially obtained from the 33 enrolled patients. Briefly, a total of 2.5 × 10^5^ PBMCs/well were plated in IFNγ-pre-coated 96-well plates (Mabtech, Nacka Strand, Sweden) and stimulated for 18 h with the 10 different peptide pools individually, at a final concentration of 2 μg/mL each peptide. Unstimulated PBMCs were used as negative controls, whereas anti-CD3 antibody and CEF (Cytomegalovirus, Epstein–Barr virus, and influenza virus) peptide pool were added as positive controls. The number of Spot Forming Cells (SFCs) per well was evaluated using an automated ELISpot counter (ASTOR, Mabtech, Nacka Strand, Sweden). All test conditions were carried out in triplicate and the number of responsive cells towards individual peptide pools was expressed as follows: (mean SFCs in the antigen-stimulated wells − mean SFCs in the negative control wells)/10^6^ cells. Technically, responses were considered detectable (positive) when the number of SFCs/10^6^ cells in antigen-stimulated wells was 2-fold higher than in negative control wells and reached at least 60 SFCs/10^6^ cells, in accordance with the Cancer Immunoguiding Program guidelines [36].

In this study, such deca-antigenic test has been named *Multi-antigenic Myeloma-specific (MaMs)* T-cell assay, using the value of 100 SFCs/10^6^ cells as the best cut-off for clinical correlations, according to Section 2.6. (see below).

### 2.4. Cytokine Secretion Assay (CSA)

A flow cytometry-based, phenotypic, and functional characterization of IFNγ-producing MM-specific T cells was carried out in two selected patients with robust responses in the ELISpot assay, by applying the CSA Detection Kit (Miltenyi Biotec, Bologna, Italy), according to the manufacturer’s instructions and protocols we previously described [34,35], using, as specific antigenic stimulation, the 10 MMAA-derived peptide pools, individually. T-cell phenotype and memory profile of cytokine-producing lymphocytes were assessed after sample counterstaining with monoclonal antibodies for flow cytometry, including: CD3, CD8, CD4, CD62L, and CCR7 (Miltenyi Biotec, Bologna, Italy).

### 2.5. ^51^Chromium-Release Assay

The ability of the different peptide pools to elicit ex vivo cytotoxic T-lymphocyte (CTL) responses was assessed by using a standard ^51^Chromium (^51^Cr)-release assay, after 15-day ex vivo expansion of MMAA-specific CTL clones [27,28]. More in details, 5–15 × 10^6^ PBMCs collected from 3 SMM patients and 2 high-risk MGUS patients (at the time of enrollment) were cultured in vitro for 8 days, by adding a comprehensive peptide mixture representing all 10 MMAA-specific antigen pools (here referred to as “total mix”) in the absence of exogenous cytokine support. The cultures were then re-stimulated with total mix-pulsed irradiated autologous feeder cells at a responder:stimulator ratio of 1:3, in the presence of IL-2 (20 IU/mL). After two weeks, responder cells were tested in 5 h and 12 h ^51^Cr cytotoxicity assays against a panel of specific targets, including phytohemagglutinin (PHA)-stimulated autologous PBMCs (known as PHA blasts) pulsed with each MMAA-specific peptide pool individually, or with control-unrelated antigens. Target cells were treated overnight with ^51^Cr, then effector cells were incubated with 10^3^ target cells at different effector/target (E:T) ratios, ranging from 20:1 to 0.01:1. Percentages (%) of specific lysis were evaluated on the basis of ^51^Cr release.

### 2.6. Statistical Analysis

A Fisher’s exact test was applied to evaluate the significance of the association between MaMs T-cell responses and disease course. *p*-values < 0.05 were considered to be statistically significant. Receiver Operating Characteristic (ROC) analysis was performed, by using the pROC R package (version 1.16.2), to identify the best threshold of MaMs test for clinical correlations with stable vs. progressive cases and to evaluate the diagnostic accuracy of such MaMs T-cell assay, by determining sensitivity, specificity, positive predictive value (PPV), negative predictive value (NPV), and the area under the curve (AUC).

## 3. Results

### 3.1. Screening and Quantitation of MM-Specific T-Cell Responses in MGUS/SMM Patients

A total of 152 PB samples, prospectively collected from the 33 patients enrolled in the study, were analyzed by applying our multi-antigenic myeloma-specific (here referred to as “MaMs”) T-cell assay, based on IFN-γ ELISpot technology. In general, specific T-cell responses were detected against different MM-associated antigens, without clear prevalence of a distinct target, and showed a wide range in the magnitude of response (from the lowest detectable level of 60 SFCs/10^6^ cells to more than 4000 SFCs/10^6^ cells). Overall, such approach revealed the presence of anti-myeloma T-cell responses, against at least one MM-associated antigen, in 23 out of 33 (69.7%) MGUS/SMM patients during the follow-up. Focusing on MGUS patients, 15/22 (68%) were positive toward at least one antigen tested. Compared to the Mayo Clinic risk stratification, detectable T-cell responses were found, with similar rates, in 3/4 (75%) low-risk MGUS patients; 7/10 (70%) intermediate-low risk MGUS patients; and 5/8 (62.5%) intermediate-high risk MGUS patients. Among SMM patients, 8/11 (72.7%) showed detectable responses—in more details, in 2/5 (40%) low-risk SMM and 6/6 (100%) intermediate-risk SMM patients.

### 3.2. Correlation of MM-Specific T-Cell Immunity (MaMs Test) with Clinical Course in MGUS/SMM Patients

In our cohort, 26 patients were persistently classified as “stable” cases during their follow-up period, while 7 patients were defined as showing a “progressive” disease, according to the binary classification described in Section 2. In details, one MGUS patient upgraded from low to intermediate-low risk group; one patient progressed from MGUS to SMM; two patients progressed from MGUS to MM; one SMM patient stepped from low to intermediate risk; and two patients progressed from SMM to MM. 

Of note, out of 26 patients with stable disease, 17 cases (65%) showed a positive T-cell response (≥100 SFCs/10^6^ cells) against at least one MMAA, while, among 7 patients classified as progressive cases, none of them (0%) resulted positive to such MaMs test (Figure 1). Some rare T-cell responses were detected in three progressive cases, but were invariably weak, ranging 60–100 SFCs/10^6^ cells. By applying Fisher’s exact test, we found a statistically significant association (*p* < 0.05) between the positivity to the MaMs T-cell assay (i.e., ≥100 SFCs/10^6^ cells, against at least one MMAA) and the clinical course in the two groups (i.e., stable vs. progressive disease).

Furthermore, to evaluate the statistical accuracy of this MaMs test, a ROC analysis was performed, providing the best combination of sensitivity, specificity, PPV, and NPV for the test. In this manner, the statistical cut-off value of ≈100 SFCs/10^6^ cells (95, exactly) for the assessment of presence/absence of specific T-cell response to each MMAA was confirmed as the best threshold in this setting, providing in particular optimal specificity and PPV (both 100%), with lower sensitivity (60%) and poor NPV (37%), eventually generating a good AUC value (0.76) by ROC analysis.

Moreover, in Figure 2 and Figure 3, we reported in detail the serial immunological data obtained with MaMs T-cell monitoring in four representative clinical cases of monoclonal gammopathy, showing remarkable correlations between their myeloma-specific T-cell response and disease course. Notably, in the two cases showing a prolonged stable disease (pt. 12 and pt. 20, Figure 2), positive MaMs test responses to different MMAAs were invariably detected throughout the long-term clinical monitoring, although without a clear pattern in the fluctuations of the magnitudes of such responses. At opposite, the two progressive cases (pt. 31 and pt. 32, Figure 3) persistently showed absent or weak MaMs T-cell responses, with initial recovery of a myeloma-specific immunity soon after starting anti-myeloma therapy (pt. 32).

### 3.3. Phenotypic and Functional Characterization of Myeloma–Specific T Cells

In pt. 12 and pt. 20, displaying persistently positive MaMs responses, we performed a similar MaMs Cytokine Secretion Assay (CSA) coupled with multiparametric flow cytometry, to further characterize myeloma-specific IFNγ-producing T cells by assessing their surface phenotype and memory T-cell profile. Basically, we confirmed the presence of frequent IFNγ-producing T cells reactive towards different MMAAs (in particular, MELAN-A/MART-1, SLAMF7, hTERT, and DKK-1). Both CD8+ and CD4+ T cells were observed, showing both Effector Memory (CD62L−) and Central Memory (CD62L+) phenotype (Figure 4).

### 3.4. Cytotoxic Activity of Ex-Vivo Expanded Myeloma-Specific CTLs

The 15-day cultured cytotoxic T-lymphocytes (CTLs) included 77% (range, 67–92) CD3+ T cells, with 41% (range, 16–56) CD4+ T cells and 32% (range, 17–63) CD8+ T cells. CD3−/CD56+ NK cells and CD3+/CD56+ T cells were 16% (range 3–23) and 37% (range 5–58), respectively. Moreover, CTLs showed the expression of the activation marker HLA-DR (median 47%, range 26–64). In all five patients tested, the 15-day ex-vivo expanded specific CTLs (stimulated with a total antigen mixture including all the 10 MMAAs) were able to induce remarkable cytotoxic responses, with a percentage of specific lysis at least >10%, against different MMAA-expressing targets (Figure 5). In particular, CTLs obtained from pt. 22 showed the most intense (up to >40%) and broadest lytic activity (against 6/10 MMAA-expressing targets). Overall, direct cytolytic activity, mediated by both CD8+ and CD4+ T cells, was directed against PHA-blasts pulsed with JAM-1 peptide pool in four out of five patients and against DKK-1, SLAMF-7, hTERT, and SOX-2 in three out of five patients.

## 4. Discussion

Novel prognostic factors, specifically able to identify patients with progressive monoclonal gammopathy (i.e., at higher risk to develop active MM closely) are expected to remarkably contribute to the clinical management of MGUS and SMM patients. Recently, increased levels of mitochondrial DNA have been proposed for the recognition of rapidly-progressive disease in SMM patients [37]. Moreover, the pivotal role of anti-myeloma immunity in the control of PC clonal proliferation has clearly emerged during the last decades, eventually best evidenced by the impressive efficacy of novel T-cell-based immunotherapies (i.e., CAR-T cells and BiTEs) in advanced MM patients [16,22,38]. However, the identification of prognostic immunologic markers, measuring the protective anti-myeloma T-cell responses and potentially valuable to guide follow-up and immunotherapeutic decision-making, still represents an unmet clinical need in the management of patients with PC dyscrasias.

Moving from previous works demonstrating that different MM-associated antigens (MMAAs) are able to induce protective T-cell immunity, here we report a clinico-immunological prospective study of screening, characterization, and monitoring of several MM-specific T-cell responses, readily detectable in the PB of MGUS and SMM patients, during long-term follow-up. In our immunological assay, we employed a panel of 10 known MMAAs, in order to provide a comprehensive stimulation including all the myeloma antigens described so far in the literature. Such an immunomonitoring approach is aimed to identify significative T-cell profiles correlating with clinical outcome, possibly disclosing specific T-cell signatures (or dynamics) with prognostic and therapeutic relevance. Basically, this report describes a first attempt to develop a standardized (i.e., irrespective to patients’ HLAs and to patient-specific MMAA expression profiles), easy-to-perform, and feasible (such as ELISpot-based assay on PB samples) multi-antigenic myeloma-specific immunoassay (here referred to as “MaMs” T-cell assay), which could act as innovative prognostic tool, helping to predict different evolutions of monoclonal gammopathies. In particular, our MaMs test may identify patients with a protective T-cell immunity (i.e., showing positive test response, ≥100 SFCs toward ≥1 MMAA), which is invariably associated with stable disease course (AUC 0.79). This highly predictive immunological signature (PPV 100%) suggests that MaMs-based analysis could be exploited to personalize the clinico-laboratory follow-up of MGUS/SMM patients with positive MaMs response (i.e., with low immunological risk of progression to MM). On the other hand, the sensitivity of the test resulted quite low (60%), with a high probability of false negative results (NPV 37%). As a main explanation for this finding, it is likely that, although wide, the selected antigenic pool only encompasses a part of the whole spectrum of MM-associated antigens and, probably, false negative rates will drop by expanding the number of antigens tested. A further immunological reason could be that very low disease burdens (i.e., M-protein < 0.5 g/dL, which was frequently observed in lower-risk MGUS patients) are expected to induce likewise low levels of circulating T-cell responses, probably under the threshold of detection by using our ELISpot assay. Moreover, in some cases, MGUS may be related to underlying lymphoproliferative disorders, which are immunologically different from PC dyscrasias and, thus, the current MM-based antigenic pool could not be suitable for these patients. Of note, with regard to the common concept that MM patients may develop a general anergic state, our MGUS/SMM patients invariably showed—even when tested negative for MaMs responses—high levels of response towards the positive controls, included in every ELISpot assay (both anti-CD3 antibody and peptide pools derived from Cytomegalovirus, Epstein–Barr virus, and influenza virus). Thus, it seems conceivable that progressive MGUS/SMM patients may display an early selective loss of protective MM-specific immunity (possibly due to specific T-cell exhaustion mechanisms, driven by neoplastic PCs), rather than a general immunodeficiency [16,17].

However, the size of our cohort is not sufficient to provide a stronger statistical support for these opposite tendencies in positive samples between stable and progressive disease, as observed by using our binary classification model. Our work can be considered a promising proof-of-concept, possibly priming the development of new immunological prognostic tests, with potential impact in the management of MGUS/SMM patients. In perspectives, further studies are needed to (i) define the optimal MM-derived antigen pool, improving the detection of specific T-cell responses significantly associated with disease control; (ii) confirm our observations by applying broad statistical analysis in larger cohorts of MGUS/SMM patients, possibly improving the promising performance of MaMs T-cell assay; (iii) validate the prognostic role of such immunomonitoring approach in multicenter clinical studies; and (iv) try to develop a comprehensive prognostic tool, combining current models of MGUS/SMM risk stratification (based on tumor-related features) with a specific immunoassay (providing information on anti-tumor host immunity). Intriguingly, future studies could also investigate the role of MaMs T-cell assay as a complementary evaluation to minimal residual disease (MRD) assessments in MM patients [39]. Moreover, BM aspirate samples could allow the isolation and immunogenic profiling of neoplastic PCs, thus prompting to investigate whether MaMs T-cell assay can be influenced by patient-specific expression levels of different MMAAs.

As expected, in this study we did not find a clear correlation, in terms of specific targeted myeloma antigens, between MaMs ELISpot assay (based on freshly isolated PB lymphocytes) and cytotoxicity tests (using MaMs CTLs, obtained after 15-day ex-vivo cultures), probably not only because CTL expansions were carried out on few patients, as a corollary of the ELISpot-based MaMs immunomonitoring, but also because these two immunological analyses provided substantially different functional information about myeloma-specific immunity in each patient. Indeed, while MaMs-based ELISpot test promptly describes the active IFN-γ secretory profile of circulating myeloma-specific T cells, the ex-vivo expansion experiments disclosed the clonogenic potentials and killing activities of such T cells, eventually suggesting that some circulating, highly active IFN-γ secreting effector T cells could be almost terminally differentiated and, thus, less expandible in vitro. For instance, in the case of pt. 12, we observed a complete concordance of the two assays, in terms of number and type of antigens with positive response; both approaches showed a positive response against 9/10 MMAAs (i.e., all except MAGE-A1). However, the magnitudes of IFN-γ responses against SLAMF7 and BCMA antigens, which were persistently the highest during ELISpot-based monitoring in pt. 12, were not associated with a correspondent expansion of similarly responsive CTL clones. In general, to investigate clinical applications of MaMs T-cell assays, the first achievable step is to identify the relevant cut-offs for prognostic binary classifications (i.e., positive/negative response, associated with stable/progressive disease, respectively), while more quantitative evaluations and linear correlations could emerge only after standardization and diffusion of this immunological test.

While novel anti-MM immunotherapies, based on therapeutic activities of anti-myeloma T-cell immunity (i.e., CAR-T cells and BiTEs) have been successfully tested and approved for clinical use in advanced disease (i.e., relapsed/refractory MM) [40,41], no attempts have been made, so far, to investigate the rational use of specific immunotherapeutic strategies to prevent the progression from MGUS/SMM to symptomatic myeloma. To date, international guidelines only recommend a careful periodic clinico-laboratory assessment for MGUS and SMM patents, and no treatment is yet approved for SMM [1]. However, high-risk MGUS/SMM patients are expected to benefit from early low-toxicity treatments, including immunotherapeutic approaches [4,38,42,43]. Indeed, recent clinical trials have shown that early treatment of high-risk asymptomatic cases with lenalidomide-based therapy significantly delays progression to symptomatic disease and improves progression-free survival (PFS) in SMM patients [4,44,45,46,47]. Moreover, it was demonstrated that the immunomodulatory effects of lenalidomide, in particular when combined with low-dose dexamethasone, may enhance antineoplastic immunity in SMM patients, actively contributing to delayed progression to MM [48]. In this study, we also showed that ex vivo cultured T cells specific towards different MMAAs, either priming naïve T cells or expanding naturally occurring CD4+ and CD8+ memory T cells, were reactive against autologous MM-peptide-pulsed cell targets, thus suggesting that such specific cytotoxic T cells (CTLs) could represent an attractive immunotherapeutic option (namely, adoptive cell therapy, ACT), particularly for MGUS/SMM patients at higher risk of progression to MM. In this view, patient-tailored myeloma-specific CTLs could be safely tested in clinical experimental protocols of autologous ACT for MGUS/SMM patients with progressive/high-risk disease, aiming to immunologically counteract the neoplastic proliferation before the outgrowth of overt cancer. Indeed, a similar immunotherapeutic approach—based on ex-vivo expansion and re-infusions of leukemia-specific CTLs—showed an excellent safety profile and impressive antileukemic activities in patients with Philadelphia chromosome-positive acute lymphoblastic leukemia (Ph+ ALL) [49].

In perspective, it is conceivable that a putative combination of ACT approaches (i.e., CAR-T cells and specific CTLs) with anti-myeloma immunomodulatory treatments (such as lenalidomide, bispecific T-cell-engagers, and, possibly, also immune checkpoint inhibitors) could allow to taper down the detrimental cancer-induced processes of T-cell anergy and exhaustion, occurring within myeloma-specific T-cell subsets, thus leading to durable and effective immune surveillance against malignant PCs [16,38].

## 5. Conclusions

Spontaneous occurrence of antineoplastic T lymphocytes, endowed with prognostic and therapeutic potentials, has been reported in patients with different hematologic malignancies, including acute leukemias, myeloproliferative neoplasms, and multiple myeloma [16,34,50,51,52]. In this work, we contributed to shed light on emerging T-cell dynamics in MGUS/SMM patients, originally describing a new specific immunological tool (namely, MaMs T-cell assay), potentially useful to investigate the role of anti-myeloma T-cell immunity in the risk assessment of monoclonal gammopathies. Moving forward from previous observations in this field, our data well support the idea that the detection of protective MM-specific T-cell responses may represent a novel immunologic marker of effective disease control, providing the opportunity to distinguish MGUS/SMM patients with different risks of neoplastic progression and, possibly, to guide both follow-up and early therapeutic interventions.

## Figures and Tables

**Figure 1 cancers-15-00972-f001:**
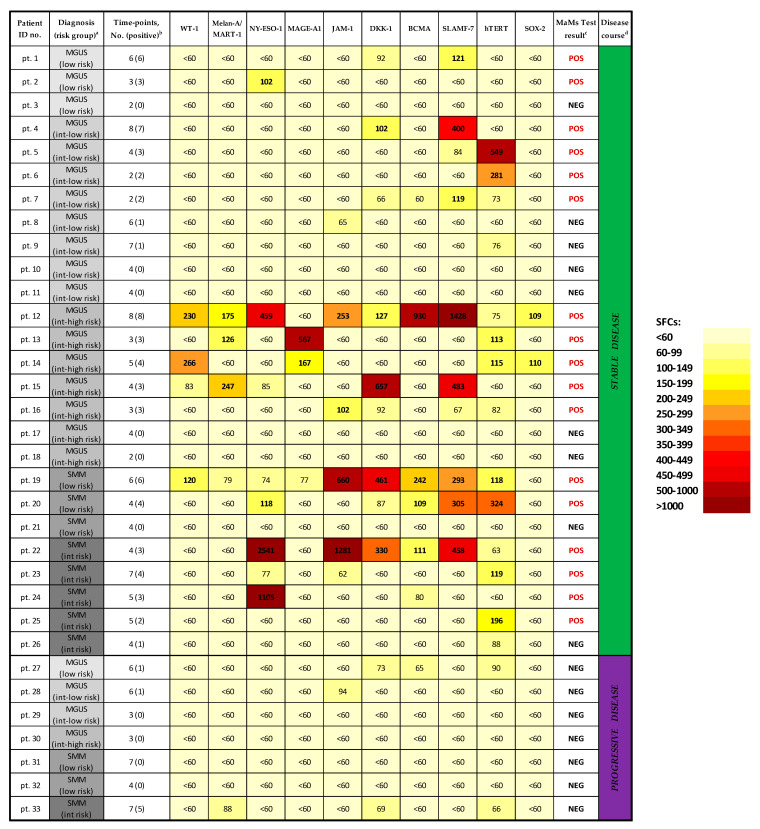
Heatmap-based report of Multi-antigenic Myeloma-specific (MaMs) T-cell monitoring in 33 MGUS/SMM patients, with either stable or progressive disease. For each patient, mean positive values (SFCs) are reported for each MM-associated antigen (MMAA) tested (values >100 SFCs are highlighted in bold). ^a^ Diagnosis and risk category (according to Mayo Clinic models) are indicated for each patient, at the time of enrollment. ^b^ Total number of follow-up samples (time-points) analyzed and number of time-points with detectable (positive) responses, according to the detection limit of the ELISpot assay (>60 SFCs), are indicated for each patient. ^c^ Overall assessment by MaMs test (either positive or negative result, according to the threshold of 100 SFCs, in ≥1 MMAA) is shown for each patient. ^d^ All patients were classified as showing either stable or progressive disease (according to the definitions reported in Section 2.1, in order to evaluate the correlation between the MaMs test result and the clinical course.

**Figure 2 cancers-15-00972-f002:**
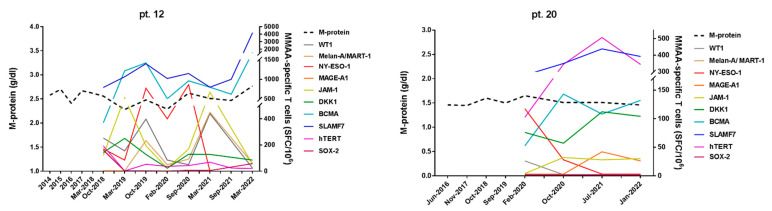
Correlation between MMAA-specific T-cell responses and disease course in two patients with stable monoclonal gammopathy. In these graphs, frequencies of IFNγ-secreting antineoplastic T cells targeting the different MMAAs (right, *y*-axis; colored continuous lines), as assessed in PB samples, at different time-points of follow-up (*x*-axis), are correlated with M-protein levels (left, *y*-axis; black dotted line). Different *y*-axis scales are applied in the graphs, according to patient’s dataset. **Left panel**: a 79-year-old woman (pt. 12), diagnosed with IgG-kappa MGUS (intermediate-high risk) in 1998, has been showing persistently elevated—but remarkably stable—M-protein levels (i.e., ranging 2.23–2.78 g/dl), throughout the last 20 years of follow-up. By analyzing several time-points over the last 4 years, we invariably detected high frequencies of different myeloma-specific T cells, responding to almost all the antigens tested (in particular, very elevated towards SLAMF7 and BCMA). **Right panel**: a 63-year-old woman (pt. 20), with low-risk IgA-kappa SMM since 2016 (BM PCs 30–40%, FLC ratio 7.85, no bone lesions), showed a fully stable disease during >5-year follow-up, with M-protein levels ranging 1.43–1.65 and no occurrence of MM-defining events. Robust myeloma-specific T-cell responses were always documented in all four consecutive tests performed between 2020 and 2022. In this case, specific T-cell responses were directed to three MMAAs (namely, SLAMF7, hTERT, and BCMA).

**Figure 3 cancers-15-00972-f003:**
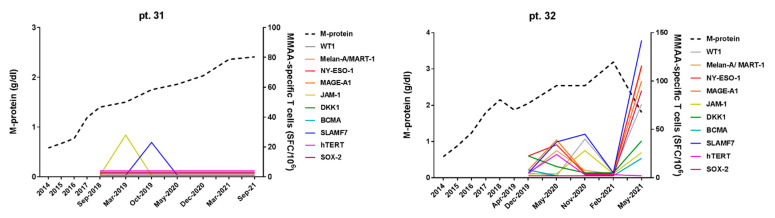
Correlation between MMAA-specific T-cell responses and disease course in two patients with progressive monoclonal gammopathy. In these graphs, frequencies of IFNγ-secreting antineoplastic T cells targeting the different MMAAs (right, *y*-axis; colored continuous lines), as assessed in PB samples, at different time-points of follow-up (*x*-axis), are correlated with M-protein levels (left, *y*-axis; black dotted line). Different *y*-axis scales are applied in the graphs, according to patient’s dataset. **Left panel**: a 45-year-old man (pt. 31), with an initial diagnosis of low-risk IgG-kappa MGUS in 2014, showed a progressive disease early, with M-protein raising from 0.58 to 1.75 g/dL in 5 years of follow-up. In November 2019, based on M-protein levels >1.5 g/dL, FLC ratio 3.27 and 15–20% of clonal PCs in the BM, in the absence of magnetic resonance imaging (MRI) lesions, a diagnosis of low risk SMM was made. In March 2021, further increments of both M-protein level (2.36 g/dL) and FLC ratio (10.4) were recorded, leading to the definition of intermediate-risk SMM. In this case, all types of MM-specific IFNγ-producing T cells were invariably negative (<60 SFCs/10^6^ cells) in several time-points analyzed, from September 2018 to March 2022 (not shown). **Right panel**: a 74-year-old woman (pt. 32), with a diagnosis of IgA-lambda SMM since November 2014, was enrolled in this study in April 2019 (showing M-protein 1.87 g/dL, FLC ratio 4.15). In February 2021, the disease frankly progressed to symptomatic MM, showing bone lesions on MRI, BM PCs >60%, and M-protein levels reaching 2.78 g/dL. Also in this case, negative (<60 SFCs/10^6^ cells) MMAA-specific T-cell responses were detected in four consecutive tests performed between 2019 and February 2021. Interestingly, in May 2021, soon after starting MM therapy, some robust and wide T-cell responses well emerged against several MMAAs (namely, SLAMF7, NY-ESO-1, and MAGE-A1), contextually with the decline of M-protein level.

**Figure 4 cancers-15-00972-f004:**
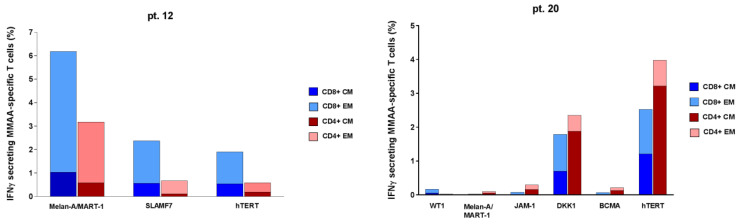
Immunophenotypic characterization and memory T-cell profile of IFNγ-secreting myeloma-specific T cells. EM = Effector Memory; CM = Central memory; MMAAs = Multiple Myeloma Associated Antigens.

**Figure 5 cancers-15-00972-f005:**
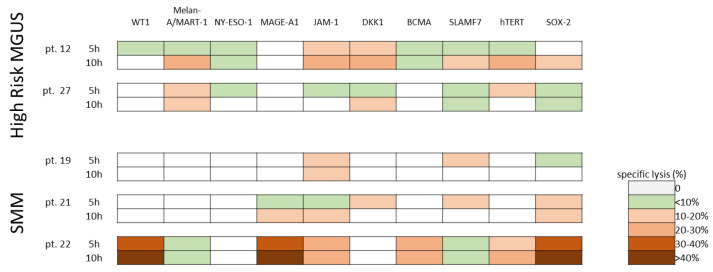
Cytolytic activity exerted by MMAA-specific CTLs. Cytotoxic activities of PBMCs from three high-risk MGUS patients and two SMM patients, after 15-day culture in the presence of peptide pools from 10 MMAAs, were tested against autologous PHA blasts pulsed with each MMAA separately. The results are represented as the percentage of specific lysis and were calculated after subtraction of background, consisting of cytotoxicity against autologous PHA blasts pulsed with control unrelated peptides. Heatmap shows the results of 5 h and 10 h cytotoxicity assays in each patient.

**Table 1 cancers-15-00972-t001:** Clinical characteristics of MGUS/SMM patients enrolled in this study. Risk categories are defined according to Mayo Clinic models.

Parameter	Value
Number of patients (Sex)	33 (15 F, 18 M)
Age at enrollment (years), median (range)	65 (26–85)
MGUS patients, total number	22
Low Risk	4
Intermediate-Low Risk	10
Intermediate-High Risk	8
SMM patients, total number	11
Low Risk	4
Intermediate Risk	7
Follow-up (months), median (range)	28 (6–44)
Number of time-points/patient, mean (range)	4, 6 (2–8)
Tested samples, total number	152

## Data Availability

The clinico-laboratory data of this study are available on reasonable request from the corresponding authors, according to privacy and ethical restrictions.

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
