# Peer review of "Prognostic Relevance of Multi-Antigenic Myeloma-Specific T-Cell Assay in Patients with Monoclonal Gammopathies"

_cancers, 2023, doi:10.3390/cancers15030972_

Round 1

Reviewer 1 Report

Lagreca et al demonstrated the correlation of Multiple Myeloma (MM)-specific T cell response against ten different MM-associated antigens and prognosis of the patients with Monoclonal Gammopathy of Undetermined Significance (MGUS) and Smoldering Multiple Myeloma (SMM). They also suggested the probability of MM-specific T cell assay as a valuable prognostic tool in the monitoring of MGUS/SMM patients. The study is interesting, but some points should be clarified.

1.     There seems a tendency in the ELISPOT positives between SD and PD. However, the number of data is likely insufficient to predict the important factors important. They need to discuss carefully.

2.     Please show the data of MM-associated antigen expression profile of myeloma cells in each patient.

3.     According to the protocol of ELISPOT assay, did they use freshly isolated PBMCs or frozen one? They need to describe it.

4.     The summary of Multi-antigenic Myeloma-specific (MaMs) T-cell monitoring in patients was shown in Table 2. However, data of persistent T-cell responses are missing. It was not clearly supported by the documentation in lines 251-252, “Notably, by using this approach, the striking result is that all the patients with a stable disease showed positive, robust (≥100 SFCs/106 cells) and persistent (≥2 consecutive positive samples) T-cell responses, against at least one MMAA.” What did “data summary” and persistent mean ? What time points of the number of ELISPOT were shown? That is, it is unclear whether the score indicated the average of 6-8 time-points or one of representative. Authors should describe them carefully this in Materials and Methods.

5.     The author showed the monitoring data of MMAA-specific T cell response in the patients with stable disease (pt 12 and pt20) in Fig.1. This data is intriguingly. Please discuss the fluctuation of antigen response during the clinical course.

6.     According to the cytolytic activity of MMAA-specific T cells (Fig. 4), the author performed the expansion of T cells for total 14 days. Therefore, this assay is not a “short-term expansion” (line 337). They should revise this. In addition, when did they do the CTL assay ? 

7.     Comparing the killing activity with the IFN-γ response, it seems that IFN-g SFCs by ELISPOT assay does not necessarily correlate the killing activity. For example, in Pt.12, IFN-g SFCs against SLAM7 and BCMA are high from the beginning and even throughout the course. Despite, the CTLs showed the higher killing activity against JAM-1 and DKK1 compared to SLAM7 and BCMA. Please explain about the discrepancy and discuss it.

8.     There is a typo error. HL-DR should be HLA-DR (line 336 at page 9).

Reviewer 2 Report

The manuscript written by Dr., Lagreca and colleagues have developed a novel multi-antigenic myeloma specific T cell assay to evaluate the T cell responses towards different MM antigens. The authors have reported this assay can be applied to monitor the prognosis of patients with MGUS or SMM. 

Though the design idea is good, I have some questions below. 

1. For the selection of antigens. Some antigens are the specific surface antigen in MM(BCMA, SLAMF7 ), some are not so specific and common used in MM (WT1, SOX-2), what’s your criteria to choose? Please explain in detail in discussion part. 

2. The expression levels of antigen are different, will it affect the response result of each antigen, is it better to use different cutoff due to the intensity of antigen expression? 

3.How can you explain your model failed in NDMM, 2/7 positive shown in your data? 

4. In your stable disease pts, pt 3,9,10,17,18,21 didn’t show any positive response, it seems that they are not consistent with your conclusion.

5. Why do you label pt.12 and pt.20 the same two patients which is so confusing? (Line 319).

Reviewer 3 Report

In this manuscript Lagreca et al explored the predictive value of a deca-antigenic myeloma-specific immunoassay to predict progression to multiple myeloma (MM), independently from the Mayo Clinic risk score.

The study is based on solid clinical and scientific background but has some weaknesses that need to be addressed before publication.

1) The study explored the MaMs Tcell assay predictive value to predict MGUS and SMM progression. However, 7 MM cases were also included. These cases do not contribute to the analysis and should be better considered as controls.

2) Following the previous comments, authors should speculate around the general concept that MM cases may display a not-specific anergic state responsible for the low/absent T-cell response (rather than a specific anti-MM response deficiency). Should additional assays be added in future studies to rule out this possibility? Are previous studies already available addressing this issue?

3) Progressive disease is defined in the Results (line 242-243). This should be mentioned in the method section, instead.

4) "All patients were risk stratified according to the Mayo-Clinic prognostic models..." (line 146). This sentence is not correct, as this cannot be applied to MM cases.

5) “A total of 152 PB samples, prospectively collected from the 40 patients enrolled in the 223

Study” (line 223). Please, specify median/range of timepoints/patient (after excluding MM cases).

6) The document may contain some instances of high similarity with published/Internet content, please check the manuscript and, if confirmed, revise it accordingly to avoid plagiarism.

7) “… a new specific immunological tool useful to assess the individual risk of a close progression to active MM... (line 437). This information is not correct. First because the progression was defined as an increased score in Mayo-risk. Second because this evidence is not solid (33 cases - only 7 progressive event); the sample size considerably limits the statistical power. Overall, the results suggest that MaMs T-cell assay can be further explored in future and larger studies. If the authors want to assess the individual risk of a progression to active MM, they need to carry out a Cox regression analysis comparing the performance of Mayo score vs Mayo+MaMs Tcell assay and support that combination of the two perform better that the Mayo score alone. If sample size is too small to run this analysis, the authors should acknowledge their limitations and use the evidence presented in this paper as a proof of concept for future larger studies.

Round 2

Reviewer 1 Report

The authors responded well to all the questions and comments. Finally, this paper has been improved. 

Reviewer 2 Report

Dear authors,

Thanks for giving me the opportunity to review this work.There are some problems from the experiments design and data presentation. Though the authors have tried to solve it when we pointed out, they still can not show promising data. 

Many thanks.